# Conditional Knockout of Hypoxia-Inducible Factor 1-Alpha in Tumor-Infiltrating Neutrophils Protects against Pancreatic Ductal Adenocarcinoma

**DOI:** 10.3390/ijms24010753

**Published:** 2023-01-01

**Authors:** Je Lin Sieow, Hweixian Leong Penny, Sin Yee Gun, Ling Qiao Tan, Kaibo Duan, Joe Poh Sheng Yeong, Angela Pang, Diana Lim, Han Chong Toh, Tony Kiat Hon Lim, Edgar Engleman, Olaf Rotzschke, Lai Guan Ng, Jinmiao Chen, Suet Mien Tan, Siew Cheng Wong

**Affiliations:** 1Singapore Immunology Network, Agency for Science, Technology and Research (A*STAR), Singapore 138648, Singapore; 2School of Biological Sciences, Nanyang Technological University, Singapore 637551, Singapore; 3Department of Anatomical Pathology, Singapore General Hospital, Singapore 169856, Singapore; 4Department of Haematology-Oncology, National University Cancer Institute, Singapore 119228, Singapore; 5Department of Pathology, National University Health System, Singapore 119074, Singapore; 6Department of Oncology, National Cancer Centre, Singapore 169610, Singapore; 7Department of Pathology, Stanford University School of Medicine, Stanford, CA 94305, USA

**Keywords:** neutrophils, pancreatic cancer, hypoxia-inducible factor 1-alpha

## Abstract

Large numbers of neutrophils infiltrate tumors and comprise a notable component of the inflammatory tumor microenvironment. While it is established that tumor cells exhibit the Warburg effect for energy production, the contribution of the neutrophil metabolic state to tumorigenesis is unknown. Here, we investigated whether neutrophil infiltration and metabolic status promotes tumor progression in an orthotopic mouse model of pancreatic ductal adenocarcinoma (PDAC). We observed a large increase in the proportion of neutrophils in the blood and tumor upon orthotopic transplantation. Intriguingly, these tumor-infiltrating neutrophils up-regulated glycolytic factors and hypoxia-inducible factor 1-alpha (HIF-1α) expression compared to neutrophils from the bone marrow and blood of the same mouse. This enhanced glycolytic signature was also observed in human PDAC tissue samples. Strikingly, neutrophil-specific deletion of HIF-1α (HIF-1αΔNφ) significantly reduced tumor burden and improved overall survival in orthotopic transplanted mice, by converting the pro-tumorigenic neutrophil phenotype to an anti-tumorigenic phenotype. This outcome was associated with elevated reactive oxygen species production and activated natural killer cells and CD8+ cytotoxic T cells compared to littermate control mice. These data suggest a role for HIF-1α in neutrophil metabolism, which could be exploited as a target for metabolic modulation in cancer.

## 1. Introduction

Pancreatic ductal adenocarcinoma (PDAC) is a highly aggressive and comparably treatment-resistant pancreatic exocrine neoplasm [1]. Affected patients typically present with widespread metastases at the point of diagnosis, thus resulting in a poor prognosis [2]. Pancreatic cancer is the fourth leading cause of cancer-associated deaths worldwide with a 5-year survival rate of only 3–5%. Tumor cells undergo increased aerobic glycolysis compared to healthy cells and release tumor-derived metabolites, such as lactic acid, into the tumor microenvironment [3]. Pancreatic tumor cells specifically over-express glycolytic enzymes, such as hexokinase 2 (HK2), pyruvate dehydrogenase kinase isozyme 1 (PDK1) and lactate dehydrogenase A and B (LDHA, LDHB) compared to normal pancreatic cells [4]. Inevitably, ~80% of all pancreatic cancer patients are diagnosed with diabetes, and new-onset diabetes is thus recognized as an early manifestation of pancreatic cancer [5].

Myeloid cells constitute a large cellular niche in the tumor microenvironment and have critical roles in tumor progression and metastasis [6,7]. Conceptual progress over the past decade has revealed that, similar to tumor cells, these immune cells exhibit specific metabolic profiles that influence their immunological functions [8]. Thus, with immune cells playing an outsized role in modulating the tumor microenvironment, it is important to understand whether changes in the metabolic fate of these cells regulate their anti-tumoral or pro-tumoral properties. For example, we recently reported that inhibiting glycolysis in PDAC tumor-conditioned macrophages in vitro, using a competitive HK2 inhibitor (2-deoxyglucose), reversed their pro-metastatic phenotype [9]. Additionally, using an orthotopic mouse model of PDAC, we demonstrated that macrophages from tumor-bearing mice exhibited elevated glycolysis, and macrophage-specific deletion of glucose transporter 1 (GLUT1) significantly reduced tumor burden [10].

Large numbers of neutrophils infiltrate PDAC lesions, but little is known about their function and the underlying immunological mechanisms that promote tumor progression. Although originally perceived as short-lived, terminally differentiated effector cells, neutrophils are now being investigated for their plasticity and participation in cancer progression. Neutrophils have been shown to be important active players in pancreatic cancer progression [11]. For example, the infiltration of neutrophils has been deemed an indicator for distant metastasis as they promote the metastasis of circulating pancreatic tumor cells [12]. In addition, higher levels of macrophages and neutrophils in the tumor microenvironment were significantly associated with a lower survival rate of PDAC [13].

Neutrophils are glycolytic in nature and are strongly committed to anaerobic glycolysis for energy production [14] to enable its effector functions such as respiratory burst and chemotaxis [15]. Given the preferential use of anaerobic glycolysis as a means of generating energy for cellular activities, it is not surprising that neutrophils are well adapted to function in a hypoxic tumor environment where the oxygen and glucose supply is limited. This may be due to glycogen accumulation in neutrophils with evidence showing that inflammatory neutrophils exhibit a 10-fold increase in glycogen stores as compared to blood neutrophils. Glycogen accumulation and storage in these cells may enable them to be well suited in performing their effector function in the glucose-deficient tumor microenvironment [16].

Hypoxia-inducible factor 1-alpha (HIF-1α) activates the transcription of numerous genes to regulate a homeostatic cellular response to hypoxia [17,18]. Immune cells recruited to the hypoxic tumor microenvironment from the oxygen-rich bloodstream up-regulate HIF-1α expression. Here, HIF-1α regulates downstream immune effector functions and immune cell glycolytic capacity [18,19]. Given these key roles of HIF-1α in regulating the metabolic activity that drives tumor progression, it is critical to define the role of metabolism in directing immunity.

Cellular metabolism may be an important regulator of neutrophil effector function and differentiation. Understanding the mechanisms that underlie neutrophil metabolic status leading to a pro-tumoral phenotype may open new avenues for novel anti-tumor neutrophil reprogramming in which neutrophil homeostasis is restored and thereby limiting tumor initiation, growth and metastasis. Here, we addressed the question whether neutrophil commitment to a specific metabolic profile in PDAC influences neutrophil effector function and tumor progression.

## 2. Results

### 2.1. Pro-Tumoral Neutrophils Infiltrate the Pancreas of Tumor-Bearing Mice and Their Persistent Depletion Improves Tumor Outcomes

We first generated orthotopic PDAC tumors in mice by intra-pancreatic injection of tumor cells (1242 L) derived from KRASG12D/+; Trp53R172H/+; PDX-cre (KPC) mice [20]. These 1242 L tumor cells were transfected to stably express luciferase, which permits tracking of tumor progression over time. An increase in pancreas weight over time confirmed successful implantation and growth of tumor cells (Appendix A). Flow cytometry revealed a significant increase in Ly6G+ neutrophils in the blood (51.1% vs. 6.4%) and pancreas (25.1% vs. 1.3%) of orthotopic transplanted (OT) mice compared to sham, non-tumor bearing controls (Figure 1a, Appendix A). Furthermore, immunofluorescence (IF) staining confirmed that a large number of neutrophils infiltrated the pancreas of OT mice while little-to-no neutrophils were observed in the pancreas of sham controls (Figure 1b). 

We next aimed to determine the contribution of infiltrating neutrophils on tumor progression. The chemokine CXCR2 has been shown to mediate neutrophil migration to sites of inflammation. Consistent with previous findings [21], flow cytometric analysis indicated that neutrophils from the bone marrow and blood of OT mice expressed high levels of CXCR2 comparable to those of sham, non-tumor bearing controls. However, CXCR2 expression was downregulated on the infiltrating pancreatic neutrophils (Appendix A). To address the role of neutrophil CXCR2 expression in this model, we performed orthotopic transplantation of 1242 L cells into CXCR2−/− mice. We observed that neutrophil recruitment to the tumor was impaired compared to CXCR2+/+ WT OT controls (Figure 1c). Furthermore, these CXCR2−/− OT mice displayed a significant reduction in tumor burden at all time points (Figure 1d) and a significant extension in overall survival compared to CXCR2+/+ WT OT controls (Figure 1e). Taken together, results showed that CXCR2-dependent neutrophil infiltration is a key-contributing factor to poor disease outcomes.

On the basis of these findings, we considered the possible therapeutic potential of depleting these neutrophils in vivo using an α-Ly6G (1A8) antibody. Pancreatic neutrophils were 1A8 depleted for up to 21 days post transplantation of 1242 L cells, after which we observed a steady increase until day 28 (Figure 1f). Similar to CXCR2−/− OT mice, α-Ly6G-mediated neutrophil depletion resulted in a significant reduction in tumor burden at all time points measured post tumor transplantation compared to isotype-treated controls (Figure 1g). However, in contrast to CXCR2−/− OT mice, tumor growth recurred and accelerated between days 30 and 35 post transplantation, until the difference in pancreas weight between α-Ly6G-treated mice and controls was negligible by the day 35 timepoint (Figure 1g). The observed temporary tumor inhibition correlated with the temporary depletion of neutrophils from this model (Figure 1f). Moreover, the overall survival of α-Ly6G treated and untreated mice remained unchanged despite an initial delay in tumor progression (Figure 1h). In summary, these data suggest that Ly6G+ neutrophils are pro-tumoral, but α-Ly6G-mediated neutrophil depletion is insufficient to completely ablate neutrophil infiltration into the tumor. However, persistent neutrophil depletion in the tumor has the potential to markedly enhance survival in PDAC.

### 2.2. Pro-Tumoral Neutrophils Exhibit a Highly Glycolytic Signature and Are Adapted to a Hypoxic Tumor Microenvironment

While depleting neutrophils using α-Ly6G showed promise in delaying tumor growth kinetics, it remained unclear how tumor-infiltrating neutrophils were contributing to PDAC progression in this model. Tumor-infiltrating neutrophils are phenotypically different to those that circulate in the blood, as they significantly downregulate genes associated with cell-cytotoxicity and respiratory burst compared to naive neutrophils [22,23]. We thus sorted Dapi-CD45 + Lin(CD3/B220/NK1.1) − Ly6G + Ly6C+ neutrophils from the bone marrow, blood and pancreatic tumors of OT mice and performed RNA sequencing to decipher the molecular signature of mature neutrophils in these three anatomical compartments. RNA sequencing data is available on GEO under accession number GSE114504. Principal component analysis of the transcriptomic data identified notable differences between the three groups, suggesting that the tumor microenvironment primes tumor-infiltrating neutrophils to display distinct gene signatures compared to their counterparts (Figure 2a). To understand the biological relevance of these underlying differences in gene expression, we generated a heatmap clustering of the differentially expressed genes between the neutrophils from the three compartments and identified four distinct gene clusters (Figure 2b). Genes in cluster 1 were downregulated in tumor neutrophils and comprised genes involved in protein transport (GO:0015031), intracellular protein transport (GO:0006886), protein targeting to the vacuole (GO:0006623), nucleophagy (GO:0044804), and the adaptive immune response (GO:0002250) (Figure 2c). With regards to the latter, tumor neutrophils may modulate the adaptive immune response to indirectly enhance tumor progression (Figure 2c).

By contrast, genes in cluster 4 were upregulated in tumor neutrophils (Figure 2b) and consisted of genes involved in the positive regulation of cell division (GO:0051781), the chemokine-mediated signaling pathway (GO:0070098), the one-carbon metabolic process (GO:0006730), positive regulation of angiogenesis (GO:0045766), and positive regulation of cellular metabolic process (GO:0031325) (Figure 2c). We focused our subsequent analysis on these up-regulated genes in cluster 4, to understand how they impact biological function and lead to poor tumor outcomes. Several cellular metabolic processes were up-regulated, suggesting a change in the metabolic signature of neutrophils that infiltrate the tumor. In particular, genes associated with the glycolytic process (GO: 0006096) were all induced in tumor neutrophils despite the fact that neutrophils are glycolytic by nature [14] (Figure 2d).

Under normal oxygen conditions, immune and normal epithelial cells consume less glucose and exhibit low HIF-1α expression [17]. However, a reduction in oxygen tension results in HIF-1α activation and enhancement of glycolytic rates [17]. Concomitant with the up-regulation of genes associated with glycolysis was the induction of hypoxia-related genes in tumor neutrophils compared to bone marrow and blood neutrophils (Figure 2e). Specifically, the HIF-1α gene was down-regulated in the bone marrow as compared to the blood and pancreatic tumors of OT mice (Appendix A). To confirm that tumor neutrophils indeed exhibited a pronounced glycolytic signature as well as enhanced HIF-1α expression, we assessed the protein expression of glucose transporter 1 (GLUT1; a glycolytic marker) and HIF-1α by immunofluorescence. Following previous published data by Colom et al. and Evrard et al., the neutrophil marker S100A9 was used [24,25]. Notably, neutrophils in the pancreatic tumor of OT mice co-expressed high levels of GLUT1 and HIF-1α (Figure 2f). As expected, little-to-no neutrophil infiltration was observed in the sham non-tumor bearing mice, as indicated by the lack of S100A9 staining (Appendix A).

Taken together, tumor-infiltrating neutrophils from pancreatic OT mice exhibit a strong glycolytic profile with up-regulated HIF-1α. This distinct metabolic signature led us to hypothesize that the pronounced glycolytic profile of tumor-infiltrating neutrophils may be mediating pro-tumoral activity and PDAC progression.

### 2.3. High Influx of Tumor-Infiltrating Neutrophils Expressing HIF-1α and Glycolytic Markers in PDAC Patients

To determine if our findings in the orthotopic mouse model of PDAC could be recapitulated in human PDAC, we assessed glycolytic marker expression in formalin-fixed paraffin-embedded (FFPE) tumor sections and adjacent matched normal tissues obtained from PDAC patients. We used a sequential, multiplex staining protocol [26] (Appendix A) in which six markers were analyzed simultaneously: CD15 (neutrophil marker), pan-cytokeratin (CK; tumor marker), HIF-1α, GLUT1 and HK2 (glycolytic markers), and DAPI (nuclear marker) (Figure 3a). We quantified the expression of glycolytic markers, HIF-1α, GLUT1 and HK2 as a percentage of total neutrophils determined by CD15 + CK- cells in the tumor sections and adjacent matched normal tissues. We then assessed whether CD15+ cells expressed one or more of the glycolytic markers (e.g., HIF-1α + GLUT1 + HK2+ subset).

Similar to our mouse in vivo data, we observed a significant 6.2-fold increase in infiltrating CD15+ neutrophils in patients with PDAC (9.44%) compared with adjacent matched normal tissue sections (1.52%) (Figure 3b). Given the up-regulation of glycolytic and hypoxia genes involved in our mouse model of PDAC (Figure 2d,e), we were interested to determine whether the CD15+ neutrophils in our PDAC patients co-expressed hypoxia and glycolytic markers (HIF-1α + GLUT1 + HK2+). Indeed, PDAC tumors showed a significant increase in the proportion of HIF-1α + GLUT1 + HK2+ neutrophils (19.26%) compared to adjacent matched normal tissue (6.31%) (Figure 3c). We then wanted to determine whether co-expression of two markers is sufficient to differentiate PDAC tumor tissue from adjacent matched normal tissue and assessed the proportion of neutrophils that were positive for two glycolytic markers. Here, only HIF-1α + GLUT1+ neutrophils were proportionally increased in PDAC tumor tissue (23.21%) compared to adjacent matched normal tissue (8.17%), while the proportion of HIF-1α + HK2+ and GLUT1 + HK2+ neutrophils was similar between the two groups (Figure 3d).

Surprisingly, HIF-1α neutrophil expression alone was sufficient to distinguish between PDAC (88.97%) and adjacent normal tissue (64.79%) (Figure 3e). As nuclear translocation of HIF-1α indicates activation, we also analyzed HIF-1α cellular localization in CD15 + HIF-1α+ neutrophils. We observed a higher proportion of CD15+ neutrophils with nuclear HIF-1α+ expression in PDAC patients compared with adjacent matched normal tissue and no difference in the proportion of CD15+ neutrophils with cytoplasmic HIF-1α+ expression (Appendix A). These data suggest that HIF-1α is activated in a larger proportion of neutrophils in PDAC tumor tissue compared with adjacent matched normal tissue. The percentage of GLUT1+ neutrophils was also increased in PDAC patients (24.3%) compared with adjacent matched normal tissue (14.72%) but did not reach statistical significance (Figure 3e). Finally, there was no difference in the percentage of neutrophils expressing the glycolytic marker HK2 between PDAC and adjacent matched normal tissue (Figure 3e).

These data show that HIF-1α+ neutrophil expression alone can differentiate between PDAC tumor tissue and adjacent matched normal control tissue. Consistent with our hypothesis that highly glycolytic neutrophils lead to poorer PDAC prognosis, both human and OT mouse data show that tumor-infiltrating neutrophils express elevated levels of HIF-1α+, are highly glycolytic and thus may have a role in driving PDAC progression.

### 2.4. Neutrophil-Specific HIF-1α Deletion Significantly Attenuates Tumor Burden

MRP8 is expressed in all mature granulocytes as they exit the bone marrow and 10–20% of granulocytic and monocytic progenitors [27]. Although MRP8 expression is a highly specific marker of the myeloid lineage from bone marrow myeloid precursors to mature neutrophils, it is lost when immature myeloid cells terminally differentiate into tissue macrophages [28]. MRP8 is therefore specifically expressed on neutrophils and not macrophages.

Based on the observed upregulation of HIF-1α neutrophil expression in PDAC patient tumor tissue, we next sought to address the role of neutrophil-derived HIF-1α in directing PDAC disease outcomes in vivo. We bred HIF-1αfl/fl mice with MRP8cre+ mice to generate mice with a neutrophil-specific deletion of HIF-1α (designated as HIF-1αΔNφ). HIF-1αΔNφ mice exhibited a ~50% reduction in HIF-1α expression in pancreatic neutrophils at the gene expression level (Appendix A). To determine if neutrophil viability, function and recruitment were affected by the specific HIF-1α ablation in HIF-1αΔNφ mice, we performed a viability assessment of PDAC neutrophils via flow cytometry from both littermate control (HIF-1αfl/fl) neutrophils and HIF-1αΔNφ OT mice and observed that both groups of mice have similar viability levels in the blood (Appendix A). However, in the tumor of HIF-1αΔNφ OT mice, there was a significant drop in viability as compared to the control group (Appendix A). This was not surprising as Walmsley et al. reported that HIF-1α mediates neutrophil survival under hypoxic conditions [18]. Although there was a change in viability, the percentage of infiltrating HIF-1αΔNφ neutrophils (21.5%) was comparable with that of littermate control neutrophils (26.3%) (Figure 4a), suggesting a possibility that the recruitment of neutrophils was altered to balance the effect of HIF-1α deletion on survival. In addition, we observed that there were no significant changes in the baseline levels of neutrophils between both sham littermate controls and HIF-1αΔNφ WT mice (Appendix A). Notably, no significant changes in gene expression were observed in CCL2 and CCL3, which regulate neutrophil recruitment, between HIF-1αΔNφ and littermate control neutrophils (Figure 4b). These findings support the observation of similar percentages of neutrophils infiltrating the pancreas between HIF-1αΔNφ and littermate control mice indicating that cell viability and recruitment to the tumor was not affected by loss of HIF-1α.

To determine the effects of loss of HIF-1α in neutrophils, we assessed cellular glycolytic capacity with the functional Glycostress Seahorse assay. HIF-1αΔNφ neutrophils showed a ~50% reduction in glycolytic capacity (Figure 4c), and key glycolytic enzymes including GLUT1, fructose biphosphate aldolase A (ALDOA), lactate dehydrogenase A (LDHA), hexokinase 1 (HK1) and hexokinase 2 (HK2) were notably downregulated (Appendix A).

Bioluminescence imaging (BLI) showed that HIF-1αΔNφ mice transplanted with OT tumors had a significantly reduced tumor burden across all time points compared to littermate controls (Figure 4d). This finding was supported by a ~50% reduction in tumor weight between HIF-1αΔNφ and control mice at 29 days post tumor transplantation, when animals were sacrificed (Figure 4e,f). This dramatic delay in tumor progression in HIF-1αΔNφ mice was accompanied by significantly improved overall survival (66 days) compared with littermate controls (46 days; Figure 4g). Collectively, these data indicate that specific HIF-1α deficiency in neutrophils decreased the glycolytic capability of neutrophils, leading to reduced tumor progression and improved survival of tumor bearing mice.

### 2.5. HIF-1α^ΔNφ^ Neutrophils Exert Direct Anti-Tumoral Effects

Given that HIF-1αΔNφ mice showed reduced tumor progression kinetics, we explored how HIF-1αΔNφ neutrophils may have contributed to active anti-tumoral immunity. Reactive oxygen species (ROS) have a known role in anti-tumoral immunity [29], and one of the main neutrophil effector functions is oxidative burst [30]. We therefore hypothesized that the reduced tumor burden in HIF-1αΔNφ mice may be due, in part, to increased ROS production. Indeed, we observed that tumor-infiltrating HIF-1αΔNφ neutrophils produced higher levels of intracellular ROS compared to HIF-1α-sufficient neutrophils (Figure 5a). To determine whether the increased production of ROS correlated with enhanced neutrophil cytotoxic ability, we performed a cytotoxic assay using pancreatic neutrophils isolated from OT HIF-1αΔNφ mice and littermate controls and incubated them with the 1242 L tumor cells that were used to produce the OT model. Remarkably, we observed a significant increase in the cytolytic activity of HIF-1αΔNφ neutrophils at a 10:1 effector:target cell ratio at all time points (Figure 5b). Pre-treatment of HIF-1αΔNφ neutrophils with N -acetyl-L-cysteine (a ROS inhibitor) before the cytotoxic assay resulted in ~27% decrease in tumor cell lysis (Figure 5c).

Taken together, these data suggest that HIF-1αΔNφ neutrophils may be eliciting their anti-tumoral effects via ROS production, thereby improving the overall survival of HIF-1αΔNφ tumor bearing mice.

### 2.6. HIF-1α^ΔNφ^ Neutrophils Modulate the Adaptive Immune Response to Elicit Anti-Tumoral Effects

In addition to the direct anti-tumoral effects on neutrophils observed in HIF-1αΔNφ mice, our transcriptomic analysis revealed a down-regulation of the adaptive immune response (GO:0002250) (Figure 2c). This effect could be due to tumor neutrophils exhibiting a more immunosuppressive phenotype. Indeed, we observed that the expression of arginase-1 (Arg-1) was significantly up-regulated in tumor compared to blood and bone marrow neutrophils (Figure 6a). Arg-1 is a well-characterized immuno-suppressor, and a critical regulator of both innate and adaptive immunity [31]. In line with this, transcript analysis revealed that the expression of Arg-1 was significantly down-regulated in HIF-1αΔNφ neutrophils compared to WT OT neutrophils, suggesting that HIF-1αΔNφ neutrophils may be less immunosuppressive (Figure 6b). To address if this hypothesis was indeed the case, we assessed the production of cytokines by HIF-1αΔNφ neutrophils known to activate T-cell responses. Transcript analysis revealed an up-regulation of IL-12 and CCL5 in HIF-1αΔNφ neutrophils compared to littermate controls (Figure 6c). Consistent with the gene expression data, flow cytometry also revealed a significant up-regulation of IL-12 (Figure 6d) and TNF-α (Figure 6e) in HIF-1αΔNφ neutrophils compared to littermate controls.

### 2.7. Anti-Tumoral Effector Responses Are Supported by CD8+ T Cells and NK Cells Activation and Cytotoxicity

Given the increased IL-12 and TNF-α cytokine production in OT HIF-1αΔNφ neutrophils, we considered the possibility that delayed tumor growth in HIF-1αΔNφ mice may be supported by cytotoxic CD8+ T cell (CTL) and natural killer (NK) cell anti-tumoral effector functions. Here, we assessed lymphocyte infiltrates in the tumor according to the gating strategy, as shown in Appendix A, and observed a significant increase in the percentage of CTLs (5.2% of total CD45+ cells) and NK cells (3.7%) in HIF-1αΔNφ compared with littermate controls (3.3% and 1.4%, respectively) (Figure 7a,b). In addition, both CTLs and NK cells had elevated levels of perforin (Figure 7c), granzyme B (Figure 7d) and IFNγ (Figure 7e) in HIF-1αΔNφ mice compared to littermate controls.

These data show that, in addition to the anti-tumoral properties of HIF-1αΔNφ neutrophils, tumor progression may also be restrained, in part, by HIF-1αΔNφ neutrophils enhancing the anti-tumoral cytotoxic effect of NK and CTLs. These data show for the first time that HIF-1α expression in neutrophils during PDAC progression promotes a pro-tumoral phenotype. Conditional knock out of HIF-1α in neutrophils was able to effectively reverse the neutrophil pro-tumoral phenotype towards an anti-tumoral one to improve outcomes.

## 3. Discussion

In the year 2000, six hallmarks were proposed that characterize all cancers: self-sufficiency in growth signals, insensitivity to anti-growth signals, evading programmed cell death, limitless replicative potential, sustained angiogenesis and tissue invasion and metastasis [32]. Over the past decade however, two additional emerging hallmarks have been suggested, including reprogramming of energy metabolism and evasion of immune destruction [33]. Many studies have shown that myeloid cells constitute a prominent niche in the tumor microenvironment and have critical roles in tumor progression and metastasis [34,35]. Here, we have started to address the relevance of these two hallmarks in the context of pancreatic cancer (PDAC) by exploring how neutrophil metabolism influences tumor progression. We found that neutrophils infiltrate pancreatic tumors in large numbers and that these neutrophils exhibit a pronounced glycolytic signature and express high levels of HIF-1α. Neutrophil-specific deletion of HIF-1α significantly reduced tumor burden and converted the tumor-infiltrating neutrophil from a pro-tumorigenic to an anti-tumorigenic phenotype, which was accompanied by increased numbers and activity of activated NK cells and CTLs (Appendix A).

Neutrophils infiltrate tumors in a CXCR2-dependent manner [36] and comprise a significant proportion of infiltrating myeloid cells in cancerous tissues [37]. We confirmed this phenomenon in our pancreatic tumor model using CXCR2−/− OT mice, where we found little-to-no neutrophils infiltrate the tumor and a significantly reduced tumor burden. Although the pancreas weight was reduced by >2 fold in CXCR2−/− OT mice, the survival of these mice only improved marginally (9 days average increase in life expectancy). We hypothesize that this effect may be due in part to increased infiltration of IL-17 producing γδ T cells in CXCR2−/− OT mice as compared to controls. Indeed, it has been shown that γδ T cells are a major source of IL-17 in the cancer microenvironment, and IL-17 is a major contributor to tumor growth [38].

Given the potential for tumor inhibition observed in CXCR2−/− OT mice, we treated OT mice with an α-Ly6G antibody in a therapeutic attempt to block neutrophil recruitment. However, due to incomplete neutrophil depletion, the tumor kinetics re-accelerated in the α-Ly6G antibody depletion model, and treated mice eventually succumbed to tumor burden. This outcome is not surprising, given previous reports discussing the partial depletion of neutrophils using α-Ly6G [39,40].

Numerous studies have started to reveal the importance of immuno-metabolism in disease pathogenesis [41,42]. Activated immune cells exhibit specific metabolic profiles that modulate their downstream functions [43]. Altered cell metabolism that results in tumor initiation and growth has received renewed attention, and a functional role for HIF-1α in metastasis has been described [44]. HIF-1α has been shown to activate hypoxia-responsive genes as one of the major mediators of hypoxic response [45,46]. Under hypoxic conditions, HIF-1α can alter vascularization, angiogenesis, survival and metabolism [17,47]. In the context of cancer, HIF-1α may be a master transcriptional factor controlling cellular and developmental processes in response to the metabolic phenotype of the tumor cell [47,48]. In immune cells, HIF-1α increases the transcription of glycolytic enzymes and thus promotes the activation states of these cells and their response to hypoxia. For example, in LPS-activated macrophages [49] and Th17 T cells [50], HIF-1α signaling and aerobic glycolysis are two key features of activated immune cell status. Hypoxia leading to HIF-1α signaling has also been shown to regulate the immune suppression and the immunosuppressive function and differentiation of myeloid-derived suppressor cells (MDSCs) [51,52]. Corzo et al., reported that tumor MDSCs suppressed both antigen-specific and non-specific T cell activity with an up-regulation of Arg-1 and down-regulation of ROS. Exposure of spleen MDSCs to hypoxia resulted in the conversion of these cells to non-specific suppressors with HIF-1α being the main driver of their immunosuppressive effector functions [52].

As glycolysis is not the most effective way to generate cellular energy, we hypothesized that the increase in glycolysis observed in tumor-infiltrating neutrophils were enabling these cells to increase energy production. Our transcriptomic analysis of neutrophils from bone marrow, blood and tumors of OT mice found an up-regulation of glycolytic factors and HIF-1α expression specifically in tumor-infiltrating neutrophils and not neutrophils from the bone marrow or blood of the same mouse. Our hypothesis is consistent with studies describing the involvement of HIF-1α in glycolysis and induction of immune-cell activation [53,54]. Neutrophils are glycolytic in nature and undergo a Warburg-like shift towards glycolysis as their main means of generating energy [54]. Together with the highly hypoxic tumor microenvironment, it is not surprising that we observed a high influx of HIF-1α-expressing neutrophils into the pancreas of OT mice.

Consistent with our in vivo PDAC mouse model, we detected high neutrophil numbers in the pancreatic tumors of PDAC patients. Specifically, we detected highly glycolytic HIF-1α + GLUT1+ infiltrating neutrophils in tumor tissue. This glycolytic signature of infiltrating immune cells is not without precedence as preliminary data from our lab suggest that tumor-infiltrating macrophages also exhibit increased glycolytic marker expression. In addition, only HIF-1α expressing neutrophils were significantly increased in proportion in the tumor tissue compared to GLUT1+ and HK2+ neutrophils, highlighting that HIF-1α may have an important role in tumor progression. Indeed, neutrophils isolated from HIF-1αΔNφ mice were less glycolytic and HIF-1αΔNφ mice were significantly protected from tumor burden compared to OT littermate controls. This effect was not due to differences in neutrophil numbers recruited to the tumor site, as no difference in the percentages of neutrophils was observed in HIF-1αΔNφ as compared to control mice. This finding suggests that HIF-1α is not required for neutrophil migration into the pancreas and is consistent with the literature showing that hypoxia inducible factors are dispensable for myeloid cell tracking during inflammation and hypoxia [55]. Neutrophil recruitment has been shown to be mediated by chemoattractants secreted by neutrophils such as CCL2, CCL3, CCL19 and CCL20 [11,56]. Specifically, we analyzed the level of CCL2 and CCL3 chemokines, partly required for neutrophil recruitment and activation [57], and found similar levels between HIF-1αΔNφ and littermate HIF-1αfl/fl control mice. While chronic inflammation promotes tumor progression, tumor cells themselves also attract immune cells via chemokines such as IL-8, CCL2 (MCP-1), CCL3 (MIP-1α), CCL5 (RANTES), CXCL6 (huGCP-2), and cytokines such as IL-1β, IL-6, TNFα, GM-CSF, and G-CSF, inducing inflammation [58]. As such, future work includes further interrogating the cancer–neutrophil crosstalk and the latter’s recruitment into the tumor to understand the effect of HIF-1α in tumorigenesis.

HIF-1αΔNφ reduced the rate of glycolysis and decreased the expression of Arg-1, an immunosuppressive molecule [59]. This reduction in Arg-1 (and an increase in CCL5) is strongly indicative of a pro-inflammatory phenotype. The increase in ROS production and cytotoxic activity alongside the production of IL-12 and TNF-α cytokines is consistent with previous studies [29,60]. These characteristics of HIF-1αΔNφ neutrophils promoted the anti-tumoral activity of CTLs and NK cells. It is possible that tumor-infiltrating CTLs and NK cells in HIF-1αΔNφ mice received the appropriate stimuli to activate their anti-tumoral effector functions to control tumor growth. This theory is in line with previous studies showing that anti-tumoral neutrophils promote CTL activation by producing T-cell activating and attracting cytokines and chemokines, such as IL-12, TNF-α, VEGF and CCL3 [61]; conversely, pro-tumoral neutrophils produce large amounts of Arg-1 that would inactivate T-cell effector functions [62].

The data taken together suggest that HIF-1αΔNφ neutrophils have anti-tumoral activity whereas WT OT neutrophils expressing high levels of HIF-1α (as found in PDAC tumors) have pro-tumoral activity. Inherent flexibility in neutrophil phenotype is not a novel phenomenon. Previous studies have shown that at least two different polarized populations of tumor-infiltrating neutrophils are possible [63]. Most strikingly, we found that ablating the glycolytic signature in HIF-1αΔNφ neutrophils alone was sufficient to elicit a change in their pro-tumoral phenotype in PDAC without affecting their ability to produce cytokines and undergo chemotaxis.

HIF-1 inhibitors have been developed that act upstream of HIF-1α protein synthesis, HIF-1α stabilization and dimerization and their interactions with other proteins [64]. However, therapeutic drugs that target cancer metabolism are complicated, as illustrated by the unexpected adverse effects that contribute to the failures of multiple Phase II and Phase III clinical trials, which may or may not be due to the highly hypoxic tumor environment and/or drug toxicity [65]. HIF-1α is involved in numerous signaling pathways, thus rendering it a complex molecule to target without causing off-target effects [64]. As such, no selective HIF-1α inhibitor has yet been clinically approved. Although targeting HIF-1α has its challenges, modulating HIF-1α, or its downstream effectors, can be therapeutically advantageous, as shown in some pre-clinical tumor models [66,67]. Our data also support that altering HIF-1α expression in the large population of pro-tumoral hypoxic neutrophils may be an excellent target not yet exploited in tumor biology.

However, a number of unanswered questions remain. Given the increase in CTL and NK cell anti-tumoral activities in HIF-1αΔNφ mice, an additional mechanism worth further interrogation is the mammalian target of rapamycin (mTOR) signaling pathway in these effector cells. mTOR has been shown to be a serine and threonine kinase playing a role in nutrient availability regulating cell metabolism, growth and survival in both myeloid and lymphoid cells [68]. Is the increased CTL and NK cell function associated with an increase in mTOR activity in the tumor microenvironment? Rapamycin, an mTOR inhibitor, has been reported to impair human NK cell proliferation, cytotoxicity and granzyme B expression [69]. There may be an increase in mTOR activity as mTOR complex 1 regulates NK cell effector function via enhancing glucose uptake and promoting aerobic glycolysis. In addition, mTOR complex 1 influences CTL effector responses controlling their expression of cytolytic effector molecules [70]. Similar to NK cells, an increase in mTOR activity in tumor infiltrating CTLs in HIF-1αΔNφ mice may be expected. However, addressing these questions pertaining to the mTOR pathway in these effector cells is beyond the scope of this manuscript.

In summary, our data highlights the importance of HIF-1α expression and glycolytic status in modulating immune-cell metabolism and suggests that modulators of these metabolic pathways could be potential new targets to restrict tumor progression in PDAC. We provide a link between increased glycolytic metabolism in myeloid cells, pro-tumoral activity and PDAC progression. Targeting the immune-cell metabolic pathway may shed light on differential neutrophil functions and the contribution of these cells to PDAC progression. A better understanding of the environmental cues and factors that drive neutrophil function will have important implications for cancer immunotherapy. Future research may focus on selectively enhancing these anti-tumor neutrophil sub-populations while modulating pro-tumoral sub-populations as a novel cancer therapeutic strategy.

## 4. Materials and Methods

### 4.1. Mice

Wild-type (WT) C57BL/6, CXCR2−/−, HIF-1αfl/fl and MRP8cre+ mice were obtained from The Jackson Laboratory, bred and maintained under specific pathogen-free (SPF) conditions in the Biological Resource Centre (BRC) of A*STAR, Singapore. HIF-1αfl/fl mice (The Jackson Laboratory, Bar Harbor, ME, USA) were a kind gift from Dr. Subhra K. Biswas at the Singapore Immunology Network. Male and female mice were used, and all experimental groups were gender-matched and age-matched. HIF-1αfl/fl and MRP8cre+ mice were crossed in-house to generate progeny with HIF-1α-deficient neutrophils (HIF-1αΔNφ). All experiments were performed under the approval of the Institutional Animal Care and Use Committee (IACUC, Singapore) of the BRC, in accordance with the guidelines of the Agri-Food and Veterinary Authority (AVA, Singapore) and the National Advisory Committee for Laboratory Animal Research (NACLAR, Singapore) of Singapore (IACUC protocol #171230). In some experiments, Ly-6G (Bio X Cell; Clone 1A8, BP0075) or isotype control antibody (Bio X Cell; BP0089) was administered via intraperitoneal (i.p.) injection at 200 ug per 20 g mouse three times per week.

### 4.2. Generation of Mouse 1242 Luciferase Stable Cell Line

The luciferase gene was amplified from the PGL3 basic vector (Promega, E1751) and sub-cloned into the ITR-CAG-DEST-IRES-puro-ITR plasmid (a kind gift from Dr. Marc Schmidt-Supprian, Technische Universität München, Munich, Germany). ITR-CAG-DEST-IRES-puro-ITR (control plasmid) or ITR-CAG-Luciferase-IRES-puro-ITR (luciferase plasmid) was mixed 1:1 with SB100X transposase and transiently transfected using Lipofectamine 2000 (Thermo Fisher Scientific, 11668027, Waltham, MA, USA) into the FC1242 pancreatic tumor cell line (kind gift from Dr. Dannielle Engle, Tuveson lab, Cold Spring Harbor Laboratory, Laurel Hollow, NY, USA), derived from KRASG12D/+; Trp53R172H/+; PDX-cre (KPC) mice, as previously described [20]. Three days post transfection, transfected pancreatic tumor cells were selected using 2 µg/mL puromycin for 1 week. Cells were assayed using a Luciferase Assay kit (Promega, E1483) to confirm constitutive luciferase expression. Transfected luciferase tumor cells were designated as 1242 L.

### 4.3. Generation of Orthotopic Pancreatic Tumor Model

Orthotopic tumors were generated in mice by intrapancreatic injection of 1242 L tumor cells. Specifically, tumor cells (1 × 10^5^) were resuspended in 50 µL 1:1 solution of PBS and growth factor reduced Matrigel (Corning, 354230, Corning, NY, USA). After laparotomy, the cell mixture was injected into the body of the pancreas using a 29-guage insulin needle, to form a visible bolus. Orthotopic transplanted (OT) mice were subcutaneously administered saline, buprenorphine (10 mg/kg) and enrofloxacin (Baytril, 1.5 mg/kg) for 2 days following the surgery. Mice were euthanized 27–30 days after surgery, and tumor weights were recorded.

### 4.4. In Vivo Bioluminescent Imaging (BLI)

Tumor progression was monitored in vivo using an IVIS Spectrum Imaging System (Perkin Elmer, Waltham, MA, USA). Mice were anesthetized with isoflurane and administered 100 µL D-Luciferin (5 mg/mL; Perkin Elmer) via i.p. injection. Mice were imaged at 1, 5, 15, and 30 s exposures. The bioluminescence signal for each mouse was quantified as “Total Flux” (photons/second) using the LivingImage Software from an exposure image where the signal was not saturated. Bioluminescent images shown in the figures were at captured at 15 s exposure.

### 4.5. Immunofluorescence

For paraffin-embedded mouse tissue sections, slides were dewaxed in xylene and dehydrated through an ethanol series before antigen retrieval using Citrate pH 6 Antigen Retrieval Buffer (eBioscience, 00-4955-58, San Diego, CA, USA). Mouse sections were stained in a 4-plex staining protocol (Perkin Elmer) in the following sequence: First cycle, primary antibody GLUT1 (Thermo Fisher Scientific, RB-9052-P1), secondary antibody anti-rabbit HRP (Dako, K4003, Glostrup, Denmark), followed by OPAL FITC dye (Perkin Elmer Life Sciences) with microwave treatment using Citrate pH 6 Antigen Retrieval Buffer (eBioscience, 00-4955-58). Second cycle, primary antibody S100A9 (Abcam, 105472, Cambridge, UK), secondary antibody anti-rat HRP (Thermo Fisher Scientific, 31470), followed by OPAL Cy3 dye (Perkin Elmer Life Sciences) with microwave treatment using Tris pH 9 (Dako, S2367). Third cycle, primary antibody HIF-1α (Novus Biologicals, NB100-105, Englewood, CO, USA), secondary antibody anti-mouse HRP (Dako, K4001), followed by OPAL Cy5 dye (Perkin Elmer Life Sciences) with microwave treatment using Citrate pH 6 Antigen Retrieval Buffer. Finally, all sections were counterstained with DAPI (Sigma Life Science, D9542-5MG, St. Louis, MO, USA) and mounted. Immunofluorescence images were acquired under a confocal microscope. At least 10 images in regions of interest were taken per mouse section, and quantifications were performed using TissueQuest Analysis Software (TissueGnostics GmbH, Vienna, Austria).

### 4.6. Tissue Preparation and Dissociation

Tumors and pancreata were dissected from mice, collected in ice-cold 1 × PBS and finely minced with scissors in 15-mL scintillation vials. Enzymatic tissue digestion was performed using collagenase type IV (C5138, Sigma) and 0.5 mg/mL DNase I (Sigma, DN25) in complete DMEM (cDMEM) containing 4500 mg/L glucose and supplemented with 10% FBS (Life Technologies) and 1 × penicillin-streptomycin (PAN Biotech) for 30 min at 37 °C. A single-cell suspension was obtained by dissociating the pancreas through a 70-µm filter. Whole blood was collected by cardiac puncture and lysed in red blood cell lysis solution (155 mM ammonium chloride, 10 mM potassium bicarbonate and 0.1 mM EDTA) prior to flow cytometry staining.

### 4.7. Flow Cytometry

Single-cell suspensions of dissociated pancreata were resuspended in FACS buffer (10% fetal calf serum, 5% human serum and 0.01% sodium azide in PBS) with 0.5 mg/mL DNase I. After Fc receptor blockade with anti-FcγRIII/II (BD Biosciences, Franklin Lakes, NJ, USA), cells were stained with DAPI (Sigma Life Science, D9542), and surface stained with a myeloid or lymphoid mouse antibody panel. Myeloid cell panel consisted of: CD45 (Biolegend, 103120, San Diego, CA, USA), CD11b (Biolegend, 101239), I-A/I-E (Biolegend, 107622), Ly6G (Biolegend, 127614), Ly6C (Biolegend, 128026), CD24 (Biolegend, 101820), CD11c (Biolegend, 117318), CD64 (Biolegend, 139304), CXCR2 (Biolegend, 149604) and F4/80 (Biolegend, 123146), together with exclusion lineage markers CD3e (BD Biosciences, 553062), NK1.1 (BD Biosciences, 108706) and CD19 (BD Biosciences, 115506). Lymphoid cell panel consisted of: CD45 (BD Biosciences, 564279), CD90.2 (Biolegend, 105328), CD3e (Biolegend, 100312), CD4 (Biolegend, 100547), CD8a (eBiosciences 48-0081-82), CD25 (Biolegend, 102016), CD19 (BD Biosciences, 652291), NK1.1 (BD Biosciences, 108706), CD44 (Biolegend, 103032) and CD62L (Biolegend, 103326).

For intracellular (IC) cytokine staining, single-cell pancreata suspensions were stimulated with 100 ng/mL LPS (myeloid cell panel), or 10 ng/mL PMA and 1 µM ionomycin (lymphocyte cell panel), together with Golgi Plug (BD Biosciences, 555029) for 4 h and then stained with a myeloid or lymphoid IC antibody panel. Myeloid IC cell panel included: IL-12 (Biolegend, 560564) and TNFα (eBiosciences, 12-7321-82). Lymphoid IC cell panel included: Granzyme B (Biolegend, 515403), Perforin (eBiosciences, 12-9392-82) and IFNγ (BD Biosciences, 563854).

### 4.8. Total Reactive Oxygen Species (ROS) Measurements

Real-time ROS generation was measured using 1 µM Oxidative Stress Detection Reagent from the Total Reactive Oxygen Species Detection Kit (Enzo LifeSciences, ENZ-51010, Farmingdale, NY, USA) according to the manufacturer’s protocol. Neutrophils were incubated with the oxidative stress detection reagent for 0 and 15 min and then analyzed using a flow cytometer (BD Biosciences). Pyocyanin (100 µM) was used as a positive control to induce ROS.

### 4.9. Transcriptomics

Bone marrow, blood and tumor neutrophils were sorted based on their expression of Dapi-CD45 + Lin(CD3/B220/NK1.1) − Ly6G + Ly6C+ from a pool of 10 WT and 5 OT mice per experiment, respectively. Total RNA was isolated by double extraction, first with acid guanidinium thiocyanate-phenol-chloroform (TRIzol, Thermo Fisher Scientific, Waltham, MA, USA) and then by a Qiagen RNeasy Micro clean-up procedure (Qiagen, Hilden, Germany). All RNAs were analyzed on an Agilent Bioanalyzer (Agilent, Santa Clara, CA, USA) for quality assessment; the RNA Integrity Number (RIN) ranged from 7.8 to 10, with a median of 9.3. cDNA libraries were prepared using 2 ng total RNA and 1 uL of a 1:50,000 dilution of ERCC RNA Spike-In Controls (Ambion^®^ Thermo Fisher Scientific) using the SMARTSeq v2 protocol with the following modifications: (1) use of 20 µM TSO, (2) use of 250 pg cDNA with 1/5 reaction of Illumina Nextera XT kit (Illumina, San Diego, CA, USA). The length distribution of the cDNA libraries was monitored using a DNA High Sensitivity Reagent Kit on a Perkin Elmer Labchip GX system (Perkin Elmer, Waltham, MA, USA). All samples were subjected to an indexed PE sequencing run of 2 × 151 cycles on an Illumina HiSeq 4000 (32 samples/lane).

Paired-end reads were mapped to the Mouse GRCm38/mm10 reference genome using the STAR alignment tool. Mapped reads were summarized to the gene level using featureCounts software and the GENCODE M7 release gene annotation. Genes with read count <5 in all samples were removed from further analyses. The limma-voom pipeline was used for differentially expressed gene analysis. Differentially expressed genes were selected with Benjamini–Hochberg adjusted *p*-values < 0.05. Hierarchical clustering and principal component analysis (PCA) was performed using the log2 transformed values of fragments reads per kilobase of transcript per million mapped reads (FPKM). Heat maps were generated with row/gene-based z-score values computed from these log2 FPKM values. Gene ontology (GO) analyses were performed using the topGO Bioconductor package. All RNA-sequencing analyses were performed in R-3.3.3 (R Core Team, 2013, Vienna, Austria).

### 4.10. Seahorse Extracellular Flux Assay

The neutrophil extracellular acidification rate (ECAR, mpH/min) was analyzed using an XFe96 extracellular flux analyzer Glycolysis Stress Test Kit (Seahorse Bioscience, 102194-100, North Billerica, MA, USA), according to the manufacturer’s instructions. All media and injected reagents were adjusted to pH 7.4. Neutrophils were harvested, sorted and resuspended in ECAR medium (DMEM base (no bicarbonate) with 2 mM L-glutamine, 143 mM NaCl, and 0.5% phenol red (pH 7.35)) and seeded in the Seahorse XF-96-well plate coated with poly-l-lysine at a density of 0.3 × 10^6^ cells per well. Cells were starved and incubated at 37 °C in a non-CO_2_ incubator for 1 h prior to the start of the assay. Three baseline ECAR measurements were made before sequential injection of glycolysis drugs. The assay consisted of four stages: (1) basal (no drugs), (2) induction of glycolysis (10 mM glucose), (3) induction of maximal glycolysis (5 µM oligomycin), and (4) inhibition of glycolysis (100 mM 2DG). Glycolytic capacity was calculated by subtracting the “mean basal ECAR values” from the “mean highest ECAR values induced by oligomycin”. At least three replicates were performed for each group.

### 4.11. Cell Cytotoxicity Assay

Cell cytotoxicity was assessed using DELFIA^®^ Technology (Perkin Elmer), according to manufacturer’s protocol. Briefly, neutrophils were sorted based on their expression of Dapi-CD45+ Lin (CD3/B220/NK1.1) − Ly6G + Ly6C+ and used as effector cells. The target 1242 L tumor cells were washed once with cDMEM, and the number of cells was adjusted to 1 × 10^6^/mL. Then, 3 mL cells in cDMEM were added to 5 μL DELFIA BATDA reagent (Perkin Elmer, C136-100) and incubated for 25 min at 37 °C before washing three times prior to the assay. Effector:target cells were co-cultured at a 10:1 ratio and measured for cytotoxic activity at 2 h, 4 h and 6 h. In some experiments, effector cells were treated with 5 mM ROS inhibitor, N-acetyl-L-cysteine (Enzo Life Sciences, ENZ-51010), for 30 min at 37 °C prior to the start of assay. The background cytotoxicity level was calculated from culture medium alone without cells, the spontaneous release was calculated from target cells alone without effector cells, and the maximum release was calculated from target cells supplemented with 10 μL lysis buffer (Perkin Elmer, 4005-0010). After incubation for different times at 37 °C, 20 μL supernatant from the sample wells was transferred to a black flat-bottom plate, to which 200 μL Europium solution (Perkin Elmer, C135-100) was added and incubated for 15 min at room temperature on a shaker. The Europium signal was then measured on an EnVision 2104 Multilabel Reader (PerkinElmer, 2104-0020). The percentage of specific release was calculated as follows: (experimental release—spontaneous release)/(maximum release—spontaneous release) × 100.

### 4.12. Reverse Transcription-Quantitative Polymerase Chain Reaction (RT-qPCR)

Dapi-CD45 + Lin(CD3/B220/NK1.1) − Ly6G + Ly6C+ neutrophils were lysed in 1 mL QIAzol Lysis Reagent (Qiagen, Hilden, Germany) and then incubated with 200 μL chloroform. Total RNA was isolated from the cells using an miRNeasy Mini Kit (Qiagen) with on-column DNase I digestion using an RNase-Free DNase Set (Qiagen). DNase-treated total RNA was reverse-transcribed to cDNA using an iScript cDNA Synthesis Kit (Bio-Rad, Hercules, CA, USA). Gene expression quantifications were performed in 384-well plates on a 7900HT FAST Real-time PCR (Applied Biosystems, Waltham, MA, USA) instrument. Arg-1, IL-12, CCL5, CCL2, CCL3, GLUT1, HK1, HK2, GPI, ALDOA, LDHA, HIF-1α, and β-actin expression was determined by qPCR with KAPA Sybr FAST ABI Prism 2× qPCR Master Mix (KAPA Biosystems, Wilmington, MA, USA) using an ABI 7900HT Fast Real-Time PCR System (Applied Biosystems). All qPCR reactions were performed in triplicate, and relative gene expression values were calculated by CT method using averaged CT values and by normalizing to β-actin. Data were then plotted as fold change to the control groups. Appendix A details the qPCR primer sequences.

### 4.13. Multiplex Quantitative Immunofluorescence (IF) Staining and Analysis

Three PDAC patient cohorts were obtained from the Stanford Medical Center, National Cancer Centre Singapore (SingHealth CIRB 2012/879/B) and National Cancer Institute Singapore (DSRB 2012/00939). The Institutional Review Board of Singapore approved collection of the Singaporean patient sections. PDAC human tissue formalin-fixed paraffin-embedded sections were stained in a 6-plex staining protocol (Perkin Elmer) in the following sequence: First cycle, primary antibody anti-pan cytokeratin 1:50 (Abcam, 27988), secondary antibody anti-mouse HRP (Dako, K4001), followed by OPAL 520 (Perkin Elmer Life Sciences) with microwave treatment using Citrate pH 6 Antigen Retrieval Buffer (eBioscience, 00-4955-58). Second cycle, primary antibody anti-GLUT1 1:200 (Thermo Fisher Scientific, RB-9052-P1), secondary antibody anti-rabbit HRP (Dako, K4003), followed by OPAL 540 (Perkin Elmer Life Sciences) with microwave treatment using Citrate pH 6 Antigen Retrieval Buffer. Third cycle, primary antibody anti-HK2 1:150 (Abcam, 104836) 1:200 (Abcam, 188571), secondary antibody anti-rabbit HRP (Dako, K4003), followed by OPAL 620 (Perkin Elmer Life Sciences) with microwave treatment using Tris pH 9 (Dako, S2367). Fourth cycle, primary antibody anti-HIF-1α 1:100 (Novus Biologicals, NB100-105), secondary antibody anti-mouse HRP (Dako, K4001), followed by OPAL 650 (Perkin Elmer Life Sciences, FP1168) with microwave treatment using Citrate pH 6 Antigen Retrieval Buffer. Fifth cycle, primary antibody CD15 1:50 (Abcam, MY-5), secondary antibody anti-mouse HRP (Dako, K4001), followed by OPAL 690 (Perkin Elmer Life Sciences, FP1171) with microwave treatment using Citrate pH 6 Antigen Retrieval Buffer (eBioscience, 00-4955-58). Finally, all sections were counterstained with DAPI (Sigma Life Science, D9542-5MG) and mounted.

At least 20 multispectral images in regions of interest were captured per patient tissue section using the Vectra Workstation (Perkin Elmer). InForm Version 2.3 software (Perkin Elmer) was used to phenotype cells and quantify the protein expression levels using multispectral fluorescence analysis and spectral unmixing [71]. All analyses were based on the “Entire Cell Total Normalized Counts” and “Nucleus Total Normalized Counts” or “Cytoplasm Total Normalized Counts” generated by the InForm software. Using in-house unpublished codes, the association of cell subset co-localization expression (for example, CD15 + HIF-1α+) between matched normal and PDAC patients was analyzed. All analyses and visualizations were performed using the R statistical language (v3.3.1).

### 4.14. Statistics

Data are presented as means ± standard error of the mean (SEM) and were generated in GraphPad Prism v7.0a. Statistical significance was determined by the Student’s *t*-test (two-tailed) or by one-way ANOVA, Tukey’s multiple comparisons test, and a *p*-value < 0.05 was considered statistically significant, * *p* < 0.05, ** *p* < 0.01, *** *p* < 0.001. Survival curves in mice were calculated using Kaplan–Meier analysis, and significance in survival was determined using the log-rank test. Transcriptomic DEGs were selected using the Benjamini–Hochberg adjusted *p*-values of <0.05. For IF analysis of PDAC patient samples, a Kruskal–Wallis rank sum test was performed to examine if the composition of a cell subset (CD15 + HIF-1α+) was significantly different between any stage. A Student’s *t*-test (two-tailed) was performed to determine differences between matched normal and PDAC patients.

## Figures and Tables

**Figure 1 ijms-24-00753-f001:**
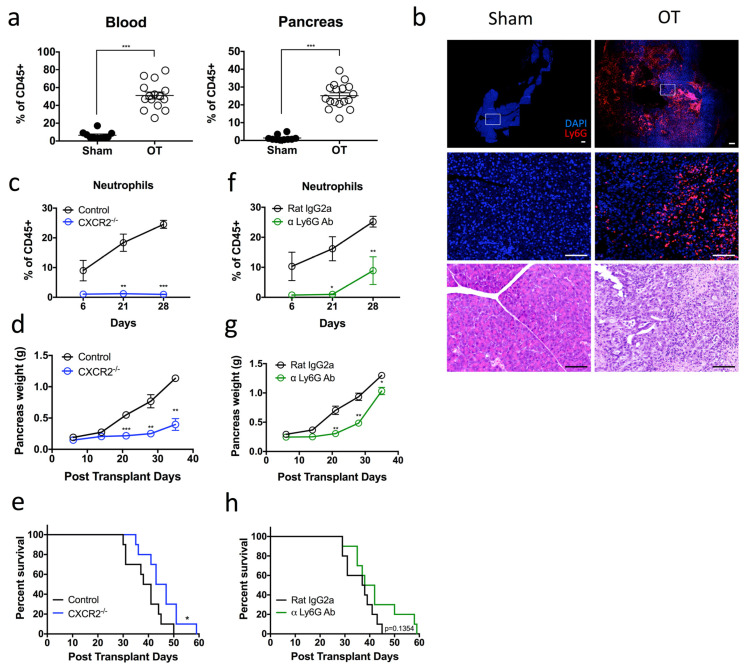
Pro-tumoral neutrophils infiltrate the pancreas of tumor-bearing mice and their persistent depletion improves tumor outcomes (**a**) Pancreatic tumors were harvested from day 28 tumor-bearing orthotopic transplanted (OT) mice, and the composition of the myeloid cell populations was assessed by flow cytometry. Scatter plot of neutrophils [Dapi-CD45 + Lin(CD3/B220/NK1.1) − Ly6G + Ly6C+] that infiltrate the pancreas are plotted as a percentage of total CD45+ cells, from OT (open circles) and age-matched sham non-tumor bearing controls (black circles). OT *n* = 15–17, sham *n* = 10; pooled from three independent experiments. (**b**) OT and control mice pancreata sections were stained for Dapi (blue) and Ly6G (red) expression. Representative immunofluorescence and H&E images of *n* = 3 experiments for each group. Scale bar = 100 µM. (**c**–**e**) Orthotopic tumors were generated in CXCR2−/− (blue circles) or control (black circles) mice. (**c**) Neutrophil infiltration into the pancreas was monitored at 6, 21 and 28 days post transplantation (CXCR2−/− *n* = 4, controls *n* = 7 per experiment for each time point). (**d**) Scatter plot of orthotopic tumor weights at 6, 14, 21, 28 and 35 days post transplantation (*n* = 3 mice per experiment for each time point). (**e**) Kaplan–Meier survival analysis of CXCR2−/− OT mice compared with control OT mice (*n* = 10). (**f**,**g**) OT mice were treated with α Ly-6G antibody (green) or isotype control antibody (black) via intraperitoneal (i.p.) injection thrice per week. (**f**) Neutrophil infiltration into the pancreas (plotted as a percentage of total CD45+ cells) was monitored at 6, 21 and 28 days post transplantation (*n* = 3–5 mice per experiment for each time point). (**g**) Scatter plot of orthotopic tumor weights at 6, 14, 21, 28 and 35 days post transplantation (*n* = 3 mice per experiment for each time point). (**h**) Kaplan–Meier survival analysis of α Ly-6G treated mice compared with isotype control treated mice (*n* = 10). Data represent the means ± SEM, * *p* < 0.05; ** *p* < 0.01; *** *p* < 0.001 by unpaired student’s *t*-test with a 95% confidence interval.

**Figure 2 ijms-24-00753-f002:**
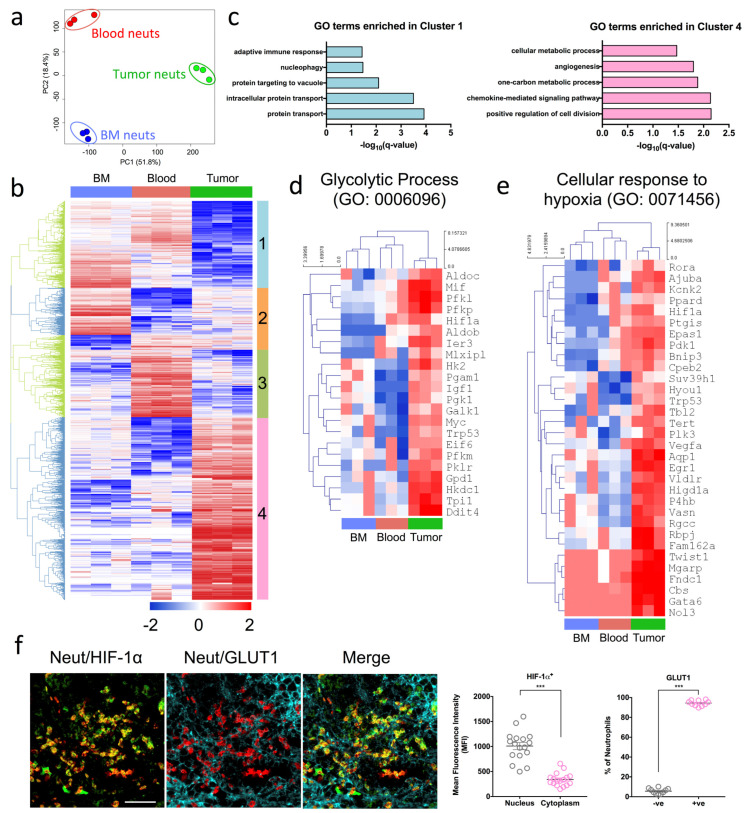
Pro-tumoral neutrophils exhibit a highly glycolytic signature and are adapted to a hypoxic tumor microenvironment (**a**–**e**) Bone marrow (BM), blood and tumor neutrophils were sorted based on their expression of Dapi-CD45 + Lin(CD3/B220/NK1.1) − Ly6G + Ly6C+ (orthotopic transplanted [OT] *n* = 5 mice pooled, three independent experiments). Hierarchical clustering and principal component analysis (PCA) were performed with log2–transformed values. (**a**) PCA plots of transcriptomic data of BM, blood and tumor neutrophils respectively. (**b**) Heat map of the differentially expressed genes selected with Benjamini–Hochberg adjusted *p*-values < 0.05. Four gene clusters (1 to 4) were manually defined and exported for gene ontology (GO) biological process analysis. (**c**) GO biological process terms enriched in clusters 1 (**left**) and 4 (**right**). (**d**,**e**) Expression of genes with row/gene-based z-score values (log2 FPKM) of cluster 4-associated genes across all sorted populations encoding (**d**) GO: 0006096—Glycolytic process and (**e**) GO: 0071456—Cellular response to hypoxia. (**f**) Representative immunofluorescence staining of pancreas sections from OT mice stained with Dapi (blue), S100A9 (red) neutrophil marker, and HIF-1α (green) and GLUT1 (cyan) glycolytic markers. Immunofluorescence images shown are representative images of *n* = 3 mice, and 10–15 fields of view. Scale bar = 100 µM. Scatter plots represent the mean fluorescence intensity (MFI) of HIF-1α+ neutrophils staining in the nucleus and cytoplasm, and percentage of GLUT1 expressing neutrophils in the tumor. Data represent the means ± SEM, *** *p* < 0.001 by unpaired Student’s *t*-test with a 95% confidence interval.

**Figure 3 ijms-24-00753-f003:**
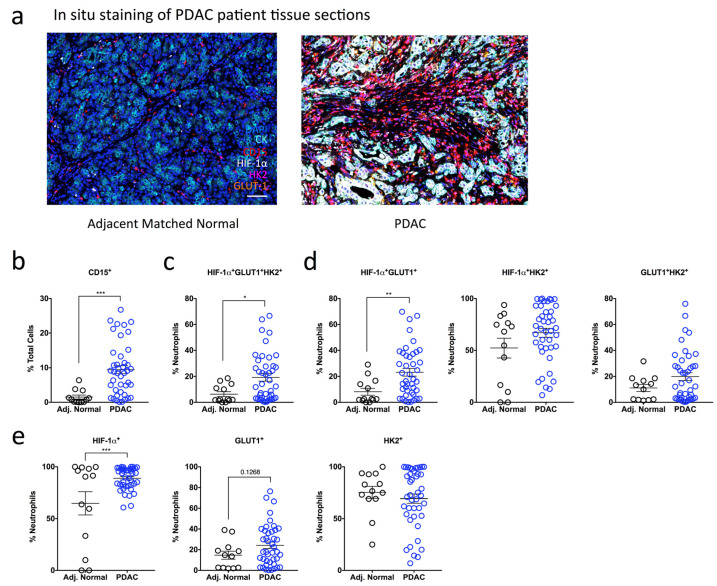
High influx of tumor-infiltrating neutrophils expressing HIF-1α and glycolytic markers in PDAC patients (**a**) A sequential, multiplex, immunofluorescence staining panel consisting of a neutrophil marker (CD15), pancreatic epithelial cell marker (CK), and metabolic markers (HIF-1α, GLUT1 and HK2) was used to stain formalin-fixed paraffin-embedded pancreatic sections obtained from multi-center patient cohorts. Images are representative of matched normal adjacent tissue (*n* = 13) and PDAC patient tumor tissue (*n* = 42). Dapi (blue), CK (cyan), CD15 (red), HIF-1α (white), HK2 (magenta), GLUT1 (orange). Scale bar = 100 µM. (**b**) Scatter plot showing the total influx of neutrophils in PDAC patient sections (blue clear) compared to matched normal controls (black clear) as a % of total Dapi+ cells. (**c**–**e**) The expression of multiple metabolic markers was quantified using Inform Software and assessed in PDAC patient tissue compared to adjacent matched normal control tissue. (**c**) Scatter plots represent the expression of triple positive HIF-1α + GLUT1 + HK2+ subset as a percentage of total CD15+ neutrophils. (**d**) Scatter plots indicate the proportions of double positive HIF-1α + GLUT1+, HIF-1α + HK2+ and GLUT1 + HK2+ neutrophils. (**e**) Scatter plots indicate the proportions of HIF-1α+, GLUT1+ and HK2+ cells as a percentage of total CD15+ neutrophils. Data represent the means ± SEM, * *p* < 0.05; ** *p* < 0.01; *** *p* < 0.001 by unpaired Student’s *t*-test with a 95% confidence interval.

**Figure 4 ijms-24-00753-f004:**
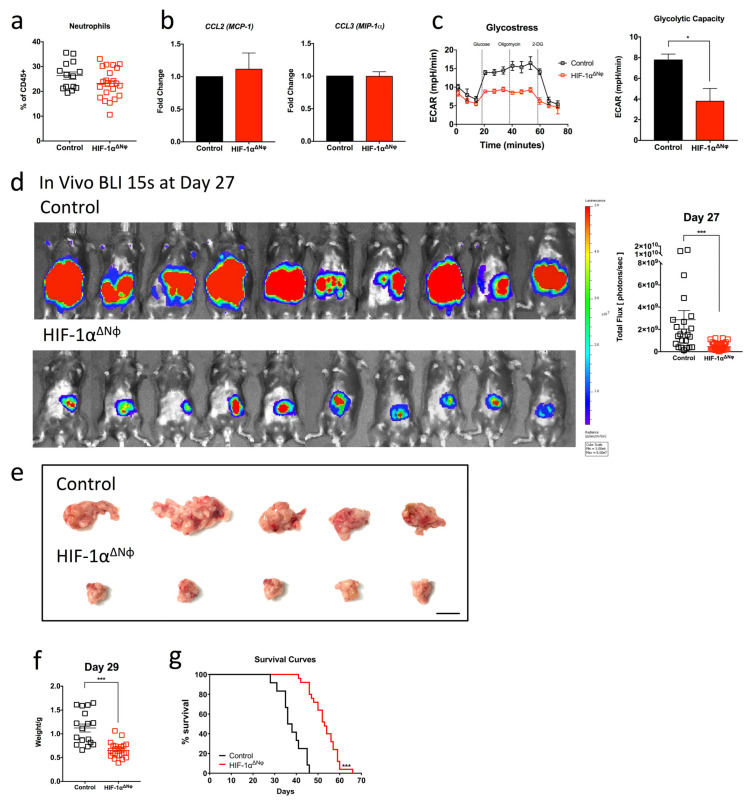
Neutrophil-specific HIF-1α deletion significantly attenuates tumor burden (**a**) Pancreatic tumors from tumor-bearing HIF-1αΔNφ orthotopic transplanted (OT) and littermate HIF-1αfl/fl control OT mice were harvested at day 28 post transplantation and the myeloid cell populations were assessed by flow cytometry. The scatter plot of infiltrating neutrophils (Dapi-CD45 + Lin(CD3/B220/NK1.1) − Ly6G + Ly6C+) are plotted as a percentage of total CD45+ cells, from HIF-1αΔNφ OT mice (red clear) and littermate HIF-1αfl/fl controls (black clear) (OT *n* = 23, sham *n* = 14; of three independent experiments, pooled). (**b**) Pancreatic neutrophils were sorted from HIF-1αΔNφ mice and littermate controls and gene expression of CCL2 and CCL3 was assessed by quantitative PCR (*n* = 3 mice, three independent experiments). (**c**) Sorted neutrophils from HIF-1αΔNφ OT and control OT mice were rested at 37 °C for 1 h and assessed for their glycolytic capacity by live metabolic flux assay. A representative Seahorse extracellular acidification rate (ECAR) trace of HIF-1αΔNφ OT neutrophils (red) compared with control OT neutrophils (black) is shown (**left**). The glycolytic capacity (**right**) was calculated from the Seahorse ECAR trace (*n* = 3–5 mice pooled per group, three independent experiments). (**d**) Orthotopic tumors from HIF-1αΔNφ OT and control OT mice were monitored in vivo over time by bioluminescent imaging (BLI; **left**). The total flux (**right**) on day 27 post transplantation was calculated from the BLI image (control, *n* = 25; HIF-1αΔNφ, *n* = 34, three independent experiments pooled). BLI images were acquired at a 15-s exposure, and the color scale was set to a lower limit of 3 × 10^6^ and an upper limit of 5 × 10^7^ photons/sec. (**e**) Representative images of HIF-1αΔNφ OT and control OT mice tumors. (**f**) Tumors were harvested on day 27 post transplantation and weighed. Scatter plot of orthotopic tumor weights from HIF-1αΔNφ OT (red clear) and age-matched OT controls (black clear) (HIF-1αΔNφ *n* = 24, controls *n* = 17; of four independent experiments pooled). (**g**) Kaplan–Meier survival analysis of HIF-1αΔNφ OT mice (*n* = 25) compared with littermate HIF-1αfl/fl control OT mice (*n* = 12). Data represent the means ± SEM, * *p* < 0.05; *** *p* < 0.001 by unpaired student’s *t*-test with a 95% confidence interval.

**Figure 5 ijms-24-00753-f005:**
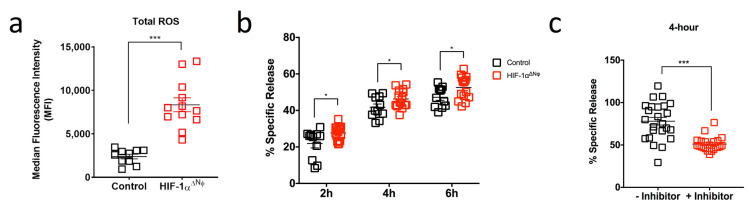
HIF-1αΔNφ neutrophils exert direct anti-tumoral effects (**a**) Scatter plot of reactive oxygen species (ROS) production (MFI) in HIF-1αΔNφ orthotopic transplanted (OT) and control OT pancreatic neutrophils (HIF-1αΔNφ *n* = 12 mice, littermate HIF-1αfl/fl controls *n* = 10; three independent experiments). (**b**) Neutrophils were sorted based on their expression of Dapi-CD45 + Lin(CD3/B220/NK1.1) − Ly6G + Ly6C+. Scatter plot shown indicates the cytotoxic capacity of HIF-1αΔNφ neutrophils at a 10:1 effector:target (E:T) cell ratio after 2, 4 and 6 h incubation with 1242 L tumor cells (*n* = 5 mice/group; *n* = 3 replicates). (**c**) HIF-1αΔNφ neutrophils were assessed for their cytotoxic capacity via ROS production. Neutrophils were treated with 5 mM N-acetyl-L-cysteine for 30 min at 37 °C. Scatter plot shown indicates the cytotoxic capacity of HIF-1αΔNφ neutrophils at a 10:1 E:T ratio after 4 h incubation with 1242 L tumor cells (*n* = 5 mice/group; *n* = 3 replicates). Data represent the means ± SEM, * *p* < 0.05; *** *p* < 0.001 by unpaired Student’s *t*-test with 95% confidence interval.

**Figure 6 ijms-24-00753-f006:**
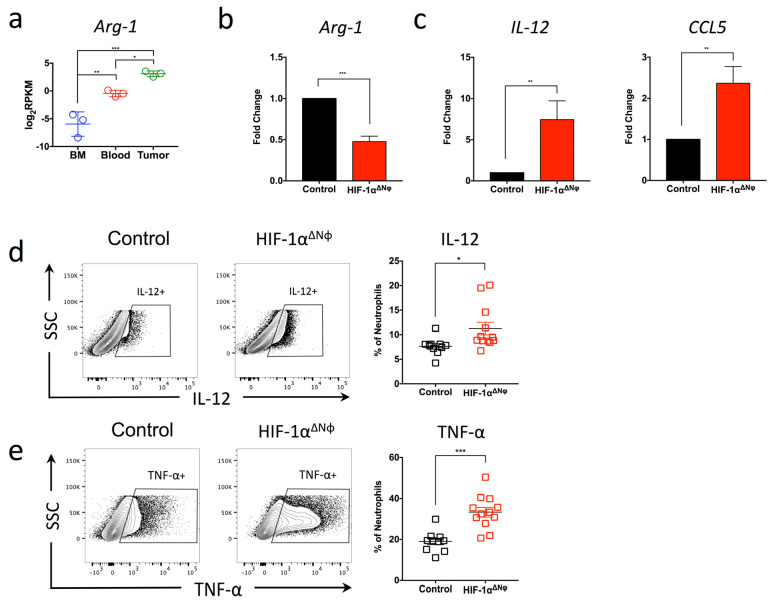
Anti-tumoral effector responses are supported by CD8+ T cells and NK cells activation and cytotoxicity (**a**) Bone marrow (BM), blood and tumor neutrophils were sorted based on their expression of Dapi-CD45 + Lin (CD3/B220/NK1.1) − Ly6G + Ly6C+ (orthotopic transplanted [OT] *n* = 5 mice pooled, three independent experiments). Gene analysis of Arg-1 was analyzed with log2-transformed values (log2RPKM). Data represent the means ± SEM, * *p* < 0.05; ** *p* < 0.01; *** *p* < 0.001 by one-way ANOVA, Tukey’s multiple comparisons test with a 95% confidence interval. (**b**) Pancreatic neutrophils were sorted from HIF-1αΔNφ and control mice, and gene expression of Ag1, (**c**) IL-12 and CCL5 was assessed by quantitative PCR (*n* = 3 mice, three independent experiments). (**d**) Representative flow cytometry plots from HIF-1αΔNφ and control neutrophils. Neutrophils were stimulated with 100 ng/mL LPS for 4 h and assessed for IL-12 and (**e**) TNF-α production by flow cytometry. Scatter plots indicate the percentage of total Ly6G+ cells, from HIF-1αΔNφ OT (red clear) and age-matched OT controls (black clear) (HIF-1αΔNφ *n* = 12, controls *n* = 10; of two independent experiments pooled). Data represent the means ± SEM, * *p* < 0.05; ** *p* < 0.01; *** *p* < 0.001 by unpaired Student’s *t*-test with 95% confidence interval.

**Figure 7 ijms-24-00753-f007:**
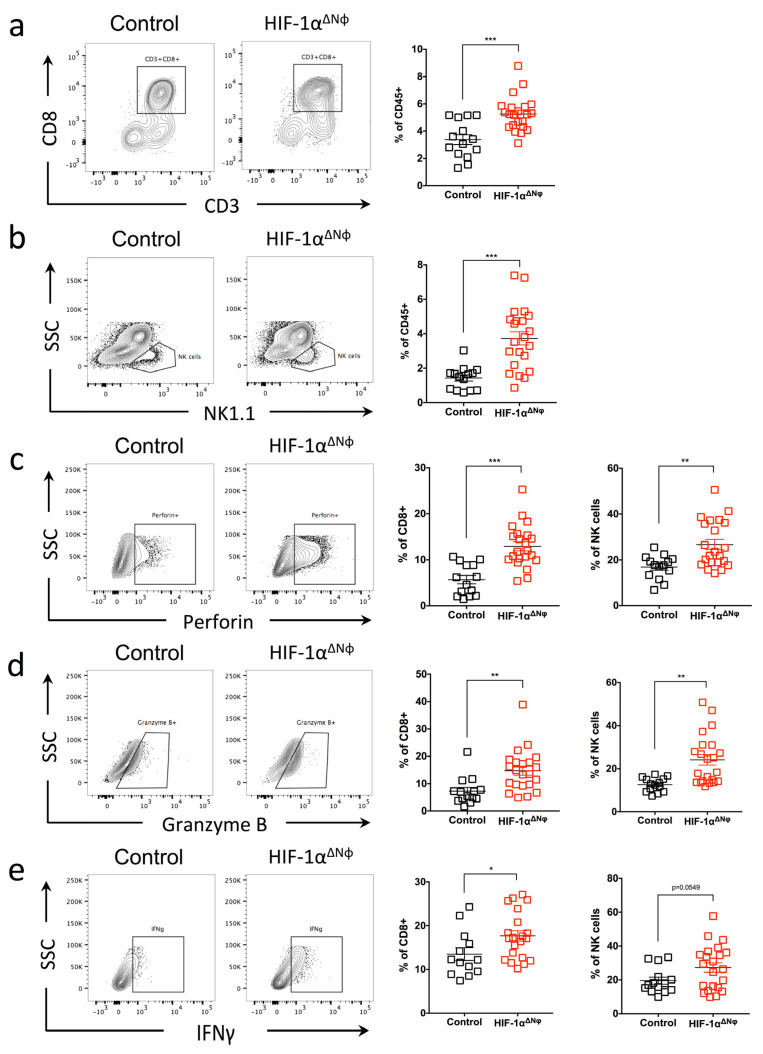
Anti-tumoral effector responses are supported by CD8+ T cells and NK cells activation and cytotoxicity. (**a**–**e**) Flow cytometry was performed in HIF-1αΔNφ orthotopic transplanted (OT) and littermate HIF-1αfl/fl control mice to assess cytotoxic T lymphocytes (CTL) and natural killer (NK) cell populations. CTL are gated based on their expression of CD45 + CD90.2 + CD3 + CD8+ and NK cells are gated based on their expression of CD45 + CD90.2-CD19-NK1.1+. Shown are scatter plots of (**a**) the % CD3 + CD8+ T cells, and (**b**) % NK cells of the total CD45+ population in the pancreata of HIF-1αΔNφ OT (red clear) and control OT (black clear) mice (HIF-1αΔNφ *n* = 22, control *n* = 14 mice per group, pooled from three independent experiments). CTL and NK cells were stimulated with 10 ng/mL PMA and 1 µM ionomycin for 4 h. Scatter plots indicate the percentage (**c**) Perforin+, (**d**) Granzyme B+, and (**e**) IFN-γ+ cells of total CTL and NK cells in the pancreata of HIF-1αΔNφ OT (red clear) and control OT (black clear) mice (HIF-1αΔNφ *n* = 22, control *n* = 14 mice per group, pooled from three independent experiments). Data represent the means ± SEM, * *p* < 0.05; ** *p* < 0.01; *** *p* < 0.001 by unpaired Student’s *t*-test with a 95% confidence interval.

## Data Availability

Data supporting the findings of this study are available within the article and its Appendix A.

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
