# Peer review of "Conditional Knockout of Hypoxia-Inducible Factor 1-Alpha in Tumor-Infiltrating Neutrophils Protects against Pancreatic Ductal Adenocarcinoma"

_ijms, 2023, doi:10.3390/ijms24010753_

Round 1

Reviewer 1 Report

The study entitled “Conditional knockout of hypoxia-inducible factor 1-alpha in tumor-infiltrating neutrophils protects against pancreatic ductal adenocarcinoma” by Siew et al.  is a very well-designed, extensive and well-written article where authors demonstrate targeting the neutrophil specific HIF1-α restricts tumor progression in PDAC.

Minor Comments:

In figure 2f, authors need to include the immunofluorescence images form control group for the better comparison.

Reviewer 2 Report

This study highlights the importance of HIF1a in tumor-infiltrating neutrophils. There are some areas where this study needs to be strengthened:

1.       In Fig.1b, labeling is missing in the immunofluorescence and H&E staining images. Which fluorescent tag is used for Ly6G?

2.       In Suppl. Fig.1d, it is appropriate to draw a graph of MFI of CXCR4 expression along with histogram overlay.

3.       Did you measure the CXCR2 ligand GCP-2/CXCL6 expression in the orthotopic PDAC tumors? Did you find any difference in the expression levels of CXCL6 in the blood, BM and tumor?

4.       When you found lower expression of CXCR2 in tumor infiltrating neutrophils, what was the hypothesis behind using CXCR2-/- mice?

5.       Did you also analyze neutrophils in other organs in CXCR2-/- mice? What was their % in blood and BM?

6.       Did you characterize these Pro-tumorigenic neutrophils in CXCR2+/+ OT and CXCR2-/- OT mice? e.g., expression levels of PD-L1, MMP9 etc.

7.       In line no# 384 “while depleting neutrophils” where? I don’t see any experiment in the figure 1, where neutrophils were depleted.

8.       Resolution of Fig.2c is poor, can be improved

9.       In Suppl Fig.2a, HIF-1a is white or green, please check all the legends carefully for any typographical errors.

10.   Why there is no GLUT1 staining in the sham non-tumor bearing control mice in suppl. Fig.2b?

11.   Again, resolution in Fig.7 is very poor and its hard to read the % in flow plots. Also include gating strategy for NK cells in suppl. Fig.

12.   NK1.1 is also expressed by some effector CD4+ T cells, NK cell analysis should be performed on live CD3-NK1.1+ cells.

Round 2

Reviewer 2 Report

The authors have addressed most of my concerns and the quality of the original manuscript has greatly improved now.